

# First automatic pH measurements in the bottom layer of the Ria de Vigo (NW Spain)

Juan L. Herrera[1], Jose González[2], Fiz F. Pérez[3], Gabriel Rosón[1], Ramiro A. Varela[1]

[1]Grupo de Oceanografía Física, Universidad de Vigo, Facultad de C.C. del Mar, Campus de Vigo, 36310 Vigo, Spain.

[2]Estación de Ciencias Mariñas de Toralla, Universidad de Vigo, Illa de Toralla s/n, 36331 Vigo, Spain.

[3]Instituto de Investigaciones Marinas, (CSIC), Eduardo Cabello, 6, 36208, Vigo, Spain.

*Correspondence to*: Ramiro A. Varela (rvarela@uvigo.es)

**Abstract.**

Oceanic Acidification is the process that describes a shift in the acid-base equilibrium caused by the rise of the $CO_2$
concentration in the ocean. The project A.RIOS lists among its goals to establish an observation network of oceanic acidification in the Rías and the Galician shelf (NW Iberia). Included in that observation network, an autonomous instrument for spectrophotometric measurements of seawater pH was deployed at the Ría de Vigo during four periods between November 2017 and May 2019. We present here the pH data for those deployments along with temperature, salinity, and pressure   data.   All   the   data   is   available   through   an   unrestricted   repository   at
https://doi.pangaea.de/10.1594/PANGAEA.909933 (Varela et al., 2019). In the author's opinion, this dataset significantly improves the temporal resolution of the pH database in the Ría of Vigo.

## 1 Introduction

Oceanic Acidification (OA) is the process that describes a shift in the acid-base equilibrium caused by the rise of the $CO_2$
concentration in the ocean, that shift is estimated to be about 0.3 pH units by 2100 (Caldeira and Wickett, 2003; Raven et al.
, 2005). OA, also known as "The other $CO_2$ problem" (Doney et al., 2009), will modify the abiotic conditions responsible for sustaining marine biodiversity. Its impact is expected to be imminent in cold environments and upwelling ecosystems such as the Rias Baixas (NW Iberia).

As part of an ongoing international effort in improving the monitoring of OA, the project A.RIOS (Acidificación de las Rías
y plataforma oceánica ibérica in Spanish; Rías and Iberian shelf acidification) lists among its goals to establish an observation network of OA in the Rías and the Galician shelf (NW Iberia) and concretely improving the temporal resolution of the pH database in the Ría de Vigo.



Included in that observation network, an autonomous instrument for spectrophotometric measurements of seawater pH was
to be moored and maintained at the Ría de Vigo (Figure 1). That instrument would provide a long-term and high-resolution
time series of pH. Such high-resolution time series of in situ pH measurements have proved to be valuable data in assessing
OA in the Mediterranean Sea (Flecha et al., 2015). As far as the authors know, this is the first attempt at obtaining automatic
in situ pH measurements in the Rías Baixas Upwelling System (RBUS).

We aim to use that instrument over long and repetitive periods through several years to capture a coherent signal of
acidification in the Ria, unmasked by other coastal processes. In this paper, we describe the details of the data acquired
during four of such deployments.

## 2 Material and methods

### 2.1 pH measurements

We used a Sunburst SAMI-pH (http://sunburstsensors.com) instrument to record the pH time series. We selected that
instrument because it was more accurate (+/- 0.003 pH units), had better stability (< 0.001 pH units/6 months), and better
precision (<0.001 pH units) than other options available at that time. The instrument is also suitable for long-term mooring.
Its pH measurement range is between 7-9 pH units and is limited to environments where salinity lies between 25 and 40.
Salinity values at the bottom of the Ría de Vigo typically fall within those ranges. Additionally, Flecha et al. (2015) used a
Sunburst SAMI-pH satisfactorily in a similar study.

The SAMI-pH measures pH using a spectrophotometric method: for each record, the instrument pumps a seawater sample
stream through the instrument. It injects the sample with 50-μL of a pH-dependent indicator solution (meta-Cresol Purple;
mCP). The SAMI-pH measures the acidic and basic forms of the indicator at peak absorbance wavelengths of 434 nm and
578 nm. For more details about the instrument's principle of operation, see Martz et al. (2003) and Seidel et al. (2008).

The SAMI-pH is an instrument reasonably simple to operate. Data is stored internally. Scheduling, flushing, and
downloading are done connecting the instrument to a PC running the manufacturer's software.

### 2.1.1. Initial test

At the beginning of the project, we compared the SAMI-pH measures with pH measures obtained using standard sampling
methods. For that, we deployed the instrument at a shallow depth during 16 days with a measurement interval of 30 min. We
collected water samples as close as possible to the instrument every 2 or 3 days using a Niskin bottle. Then, we analyzed two
replicas in the laboratory using a spectrophotometric method comparable to the one used by the SAMI-pH (Clayton and
Byrne, 1993). Figure 2 shows a comparison of the bottle data with the SAMI-pH data at times closest to the bottle



acquisitions. SAMI-pH measures are higher than the bottle measures, and there is a linear relationship between both methods. Although the relationship was not one of identity, it was consistent. Encouraged by those results, we deployed the instrument at the bottom of the Ría de Vigo.

## 2.2 The deployments in the Ría

Initially, we planned to acquire a long-term and high-resolution time series of pH by repeatedly mooring the SAMI-pH at the
same location. The instrument required periodic maintenance and data download, and was planned to recover it every 2-3 months, keeping the gaps in the time series as short as possible.

Before each deployment, we configured the SAMI-pH to take a measurement every 30 min and was set to compute pH using a constant salinity of 35. We also used a fresh battery with a nominal life of about 100 days. The reagent comes in 1L bags,
enough for 417 days. Therefore, the battery limited the deployment duration. In practice, at the 30 min sampling interval, the battery only had the energy to power the device for 60 days (Table 1). We changed the reagent bag after the second deployment.

We secured the instrument into a protective metallic cage (Figure 3). At the top of the cage, we fixed four rigid buoys to
provide positive buoyancy during the recovery phase and to maintain the cage in an upright position. We attached an acoustic release to the bottom of the cage, and a cement deadweight tied to the release served as an anchor. We designed the mooring to keep the sensor 2m above the seabed, to reduce the interference of resuspended material.

On each occasion, we lowered the instruments carefully using a crane and recorded the exact deployment location at the
moment in which the instrument landed at the bottom. After each deployment, we cleaned the SAMI-pH. Because of the high sensitivity of the pH measurements to particles housed in the internal water circuit of the SAMI-pH, we took special care when cleaning its internal water circuit, flushing deionized water through the circuit.

### 2.2.1. Mooring site A

Mooring site A is a location with a long history of oceanographic measurements in the Ría de Vigo, and its potential
synergies with other current and future research projects developed in the Ría made of it an attractive location.

We completed two deployments at that location (Table 1). Unfortunately, after the second deployment, the pH data was noisy, and we sent the instrument to the factory for reparation and recalibration. As a result, there was a gap in the time series longer than initially planned. The manufacturer pointed to the near-bottom turbidity caused by the resuspension of
sediment as a possible cause of the noise. The particles had collapsed the water circuit, and the instrument's pump required to be replaced. One possible solution at the time was to add a filter to the water intake that would prevent mud particles from



entering into the water circuit. However, there are, according to the manufacturer, conflicting reports about the filter effectivity, and we discarded that option. Since the seabed was muddy at mooring site A, we selected a new spot (mooring site B, Figure 1) with a rocky bottom for subsequent deployments.

### 2.2.2. Mooring site B

During the third and fourth deployment, we expanded our set of measures to include conductivity and pressure. Also, although the SAMI-pH records temperature (precision +/-0.01 ºC), we recorded two new temperature series using an instrument with better precision.

The conductivity, temperature and pressure series were recorded using a Seabird 37SM (calibration date: Feb 2018; temperature precision.: +/- 0.002ºC; conductivity precision: +/-0.003 mS/cm) and a Seabird 39P (calibration date: Dec 2013; temperature precision corrected using typical drift.: +/- 0.016ºC; pressure precision corrected using typical drift: +/-1.4db). The sampling interval of both instruments was 30 minutes, and we synchronized their internal clocks with the SAMI-pH internal clock.

### 2.3 Data processing

We processed the data using *R 3.6.1* (R Core Team, 2019). When available, we aligned the temperature, conductivity, and pressure data with the pH time-series. We computed salinity from the conductivity, temperature, and pressure data using the package *oce* (Kelley, D. and Richards, C., 2019); and then used the salinity to correct the pH measurements following the manufacturer instructions. Temperature, conductivity, and pressure time-series are available in the dataset.

### 2.3.1 pH quality control and smoothing

The dataset includes a cleaned version of the pH data. We applied a two steps quality control procedure. First, we rejected data outside the range 7.5-8.25. Next, we used the package *RcppRoll* (Ushey, K., 2018) to compute rolling averages (window size=25, 12.5 hrs), and then the residuals subtracting the averages from the data. Finally, we rejected points with absolute residual values greater than two times the standard deviation of the residuals. The result was the clean time series included in the data repository (Figure 4).

The dataset includes a smoothed version of the pH time series that are the rolling averages (window size=6, 3 hrs) computed using the clean data (Figure 4).

## 3 Results

Figure 5 shows the pH smoothed time series along with its trend for each deployment. We can see that deployments 1 and 2 show a higher short time variability than deployments 3 and 4. We suspect that the reason for that interference could be that bottom suspended sediments interfered with the pH measurements at position A.

Deployment 1 and 2 show positive pH trends. Also, pH correlates significantly with temperature and salinity during
deployments 3 (0.13 and 0.19 respectively; p-value <0.001) and 4 (0.66 and -0.5 respectively; p-value <0.001). Alternating upwelling/downwelling events, with the subsequent advection of low/high pH shelf waters inside the Ría de Vigo (Gago et al.; 2003a, 2003b), may account for most of the low frequency (several days) pH variability observed in Figure 5. Further analysis and discussion of the data are beyond the scope of this paper.

## 4 Data availability

pH and complimentary time-series for each deployment are available in text format in an unrestricted repository at https://doi.pangaea.de/10.1594/PANGAEA.909933 (Varela et al., 2019). We created one file per deployment.

## 5 Conclusions

We believe that this dataset, with its long high-resolution pH time series, represents a significant enhancement to the temporal resolution to the pH records that comprise the present pH database for Galician coastal waters. Among other
benefits, this dataset will help in the effort of modeling and evaluating the expected impact of OA on marine mussel cultures.

**Author contribution.** RAV, JG, and JLH conceived and designed the manuscript. JLH wrote the manuscript and prepared the dataset for publication in PANGAEA. RV and JG acquired the pH data. A.RIOS project was lead by FFP, RAV, and GR. All co-authors read the manuscript and contributed edits.


**Competing interests.** The authors declare that they have no conflict of interest.

**Acknowledgments.** This study was funded by the Spanish government through the Ministerio de Economía y Competitividad that included European FEDER funds (project CTM2016-76146-C3-3-R) and EU FEDER (CTM2016-
76146- C3-3) under the name A.RIOS in memorial of the oceanographer Aida F. Rios. We are grateful to the crews of the R/V Mytilus and Kraken for their precious help during sensor mooring and various operations. We would also like to thank Ph.D. Susana Flecha Saura from the Instituto de Ciencias del Mar de Andalucia for her advice during and after the SAMI-pH buying procedure.



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

**Figure captions**

**Figure 1: Deployment locations.**

**Figure 2: Comparison between SAMI-pH data and pH measurements from collected water samples.**

**Figure 3: SAMI-pH ready for deployment.**

**Figure 4: Quality control and smoothing of the pH time-series. Original: original SAMI-pH data. In range: data after removing points outside the pH range 7.5-8.25. QC Roll avg: Rolling averages (window size 25, 12.5 hrs) computed using data "in range." Residual: In rage data minus QC Roll avg. Clean: clean data included in the dataset. Smoothed: rolling averages (window size=6, 3 hrs) of the cleaned data.**

**Figure 5: Smoothed pH, temperature, and salinity time-series.**



| Location | Depth (m) | Deployment dates | pH series dates | temperature | conductivity | pressure |
|---|---|---|---|---|---|---|
| 42º 14.466' N 8º 45.615'W | 40 | 11/15/2017 16:00– 01/14/2018 16:00 (60 days) | 11/15/2017 16:00 – 01/14/2018 16:00(60 days) | - | - | - |
| 42º 14.466' N 8º 45.615'W | 40 | 02/02/2018 13:00 – 05/8/2018(95 days) | 02/02/2018 13:00 – 03/15/2018 9:00 (41 days) | - | - | - |
| 42º 11.253'N 8º 49.689 W | 30 | 12/11/201817:00 – 03/01/2019 (80 days) | 12/11/2018 17:00 – 02/07/2019 16:00 (58 days) | 12/11/2018 – 03/01/2019 (80 days) | 12/11/2018 – 03/01/2019 (80 days) | - |
| 42º 11.253'N 8º 49.689 W | 30 | 03/21/2019 17:00 – 05/20/2019 17:00 (60 days) | 03/21/201917:00 – 05/20/2019 17:00 (60 days) | 03/21/2019 – 05/20/2019 (60 days) | 03/21/2019 – 05/20/2019 (60 days) | 03/21/2019 – 05/20/2019 (60 days) |

Table 1. Deployment details and time-series length for each variable.






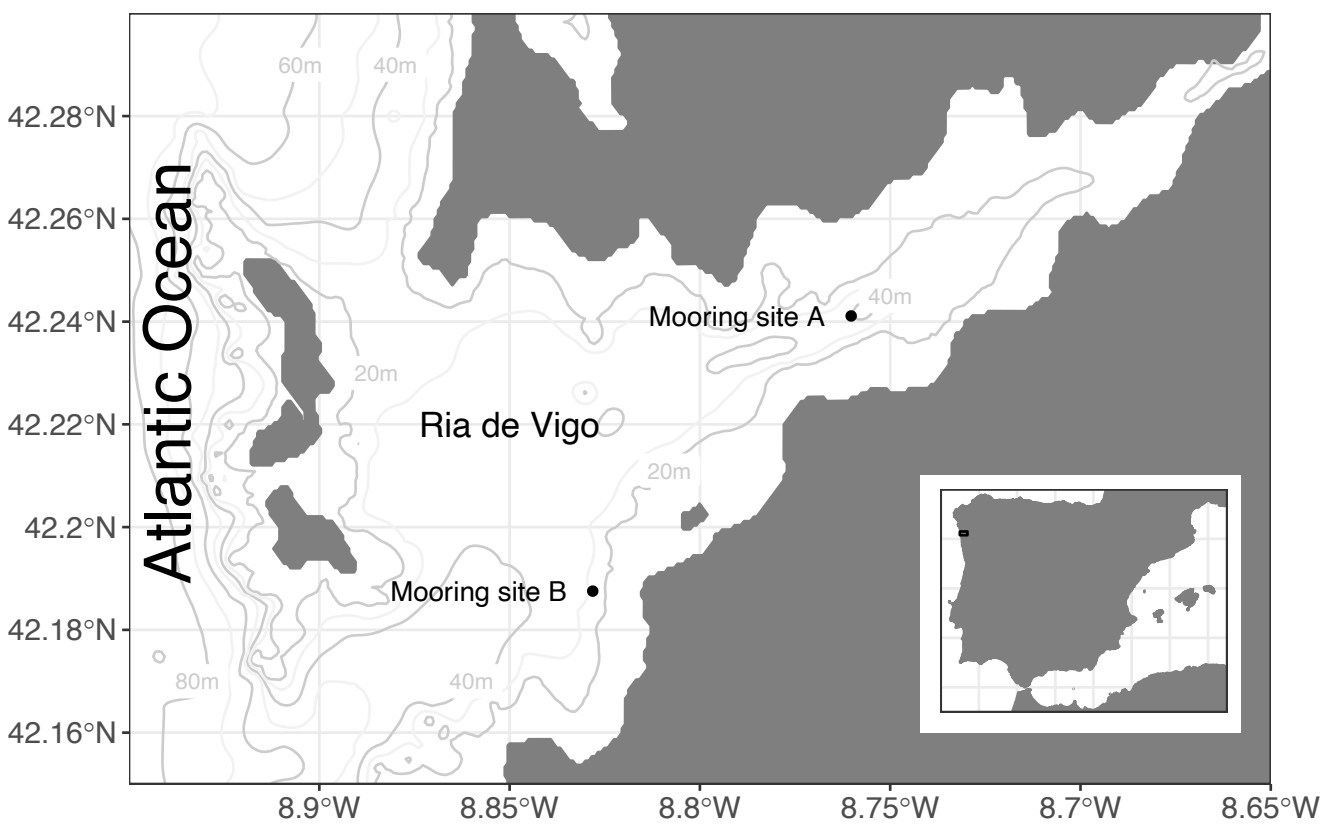

**Figure 1**





$y = 2.91 + 0.651\ x \quad R^2 = 0.94 \quad N \quad 15$

**Figure 2**



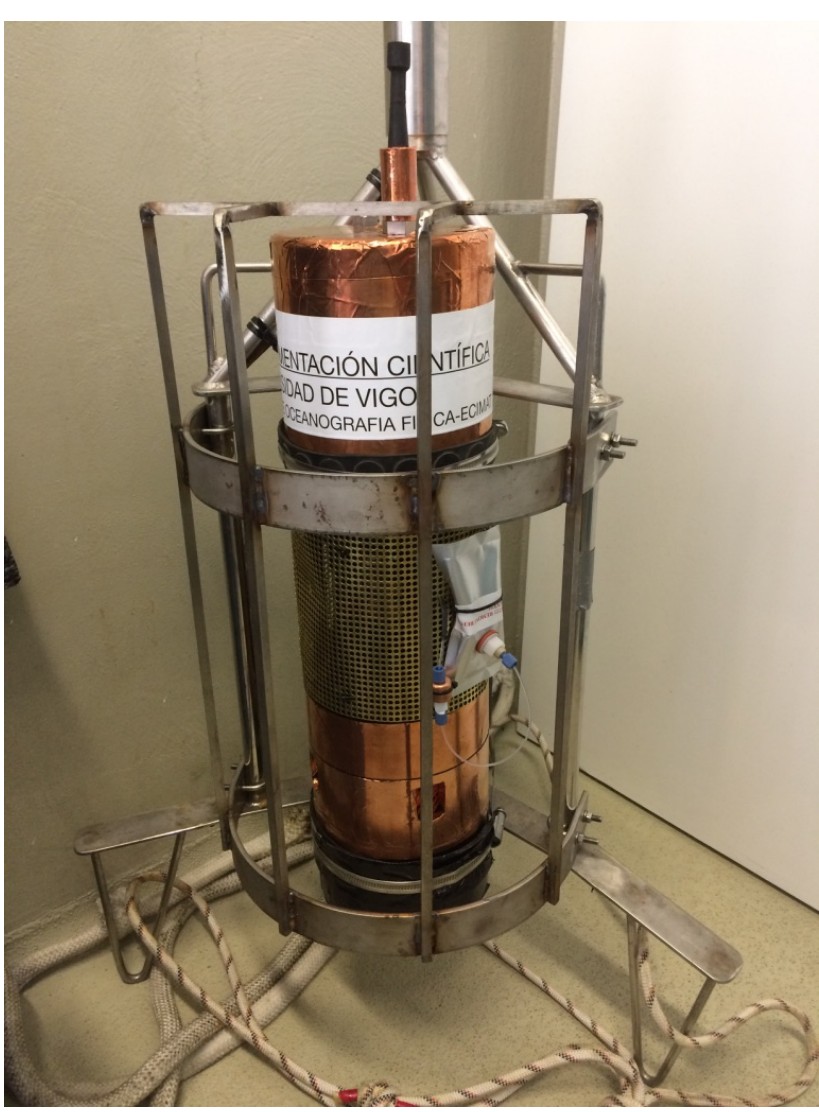


**Figure 3**






**Figure 4**





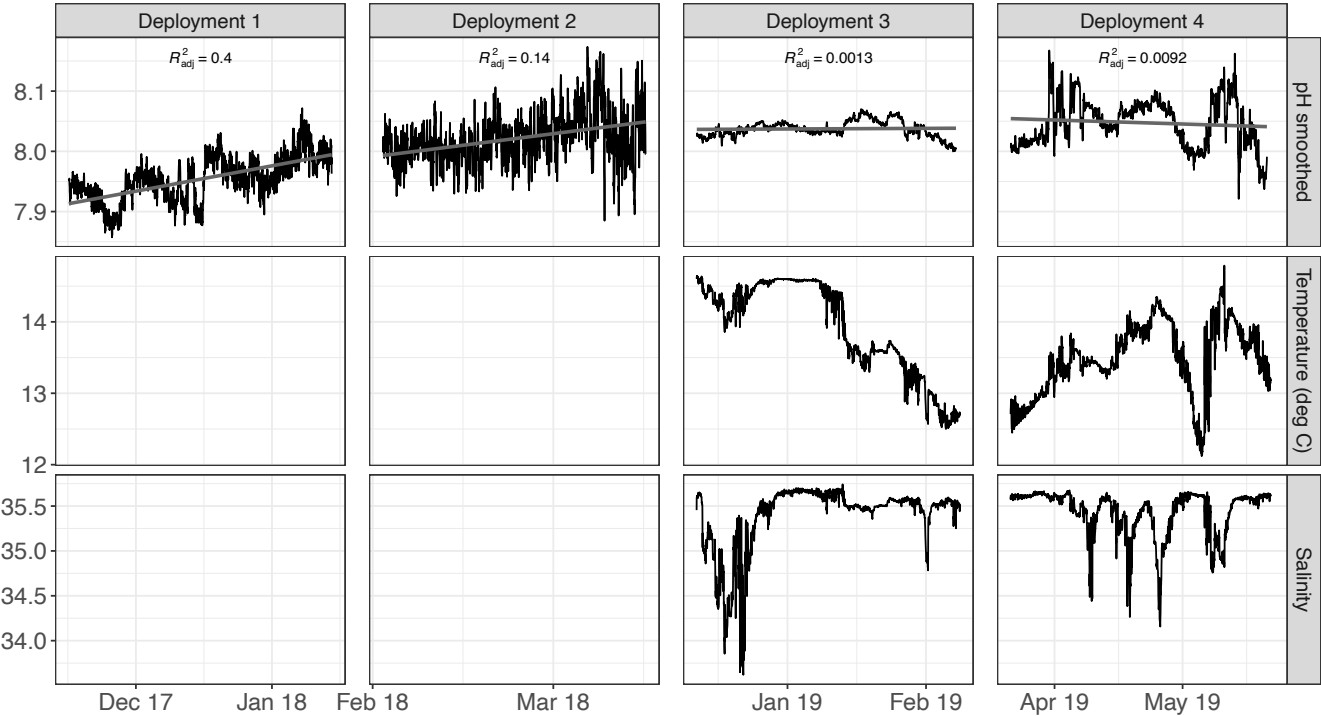


**Figure 5**