# Peer review of "First automatic pH measurements in the bottom layer of the Ria de Vigo (NW Spain)"

_Earth System Science Data, 2020_

## Referee Comment (RC1) · Michele Giani (Referee) · 7 Apr 2020

The data presented are of scientific interest in order to better understand the variability of pH at different time scale daily/seasonal in coastal waters. The ms is appropriate to support the publication of a data set, but a deeper justification of the short term pH variations observed during the deployments is considered necessary. The comparison of the pH probe with laboratory measurements indicates that the data are of good quality. The data presented consist of 4 set related to 4 different deployments carried out the first 2 in a deeper station of the Ria of Vigo and the second 2 in a shallower one less influenced by sediment resuspension. The first two data set show a high noise that according to the authors is related to the resuspended particles. The scientific relevance of the published data could be better motivated than only to "improve the

temporal resolution of the pH database in the Ria of Vigo". The authors could consider the differences among coastal and ocean definition (Klein et al., Global Change Biology 2019) also to motivate the relevance of the understanding of the processes affecting the CO2 system in coastal waters. I think that it would be useful if the authors could also evidence if there is a short term variability potentially related to tidal or light variations (depending on the local water transparency). A more sound motivation of the range chosen for the pH values check (7.5-8.25) should be provided. As the noise in Deployment 2 is much higher than in Deployment 1 and the reason according to the authors is due to the sediment resuspension can this data be still considered valid? Could the authors overlap the lab measurements on the various time series plots? Regarding the accuracy of the SAMI pH the authors refer to the nominal data reported by the factory, did they checked it? Was a purified indicator used also for the comparison between SAMI – pH and laboratory measurements? Potential errors introduced in pH estimate by using unpurified m cresol purple should be considered and discussed. The reversal of the correlation between pH and salinity should be better represented and explained than only assessing that it is attributable to alternating upwelling/downwelling events. The data related to the pH vs temperature and pH vs salinity regressions could be presented in plots and better discussed. It seems that during the Deployment 3 for colder water temperature ($<13.5°C$) the relationship between pH and T follows a positive linear relationship (r=0.885, p<0.001) whereas for temperature higher than $13.5°C$ the relationship is reversed (r=-0.583, p<0.001). The effect of the salinity on the pH it does not follow a linear relationship however a change is related to salinity: for higher salinity there seem to be an inverse relationship between pH and salinity ( r=-0.3107, p<0.001). In the Deployment 4 there is a much higher short term variability of pH (>0.1 units/day) with strong change with respect to the deployment 3 which should be better addressed, presumably it could be related to variations in the seawater characteristics. Also during this deployment the relationship of pH is inverse with salinity and is stronger at salinity higher than 35.4 (r=-0.582, p<0.001). A justification of the abrupt pH change could help to show the validity of the collected data. In the references related to

autonomous pH time series only similar studies carried out in Spain are considered, a comparison with similar approaches in other geographic areas could be useful in order to evidence the relevance of the autonomous pH monitoring for understanding coastal carbon dynamics and processes.

Minor comments: L. 55. avoid the repetition of "measures" L.124. "Deployment 1 and 2 show positive pH trends". I suggest specifying "temporal trends" Figure 1. In the small inserted figure, the study area could be better evidenced. Figure 2. The statistical significance of the linear regression could be added in the figure. Figure 4. Avoid the overlap of the numbers on Y axis Figures 4 and 5. I suggest to evidence directly in the graphs those referring to station A and those referring to station B. Table 1. The title of the columns should be similar as for the measured parameters all refer to time series: pH series dates, temperature series dates,... or could be grouped under a common title as "time series dates".

---

## Referee Comment (RC2) · Jens Daniel Mueller (Referee) · 5 Jun 2020

**1 Short summary**

The authors present pH time series data from two locations in the Vigo estuary recorded by in-situ spectrophotometric measurements. As part of an observation network, a commercially available Sunburst SAMI-pH sensor was deployed twice at each location. Prior to the actual deployments, comparison pH measurements were made on discrete samples collected during a shallow water test deployment. Each of the subsequent four deep water (30-40m) deployments covers about two months within the time frame from November 2017 to May 2019. All deployments took place from November to May. Instruments were deployed near the bottom under fully marine

conditions (salinity 33-36). Spectrophotometric pH measurements performed over a muddy seabed during deployments 1 and 2 are believed to be affected by suspended sediments. The authors apply a quality control procedure, which consists of the removal of (i) pH data outside a pre-defined pH range (7.5-8.25) and (ii) pH data that deviate more than two times the observed standard deviation from a rolling average. The full suite of temperature, salinity, and pressure data is made available only for deployment 4.

**2  General comments**

Obtaining pH data with high quality and spatio temporal resolution is an important task in order to track ocean acidification and decipher long-term trends from natural variability. The authors pursue this goal by the deployment of state-of-the-art sensor technology. In principal, the presented data set could be considered significant and unique. However, the usefulness for future interpretations of the data set in its current form is restricted, mainly due to insufficient methodological information, high uncertainty in the recorded pH data, inappropriate data processing procedures, restricted temporal coverage, and lack of additional data from the observation network. As a consequence, the data set quality does not allow to achieve the stated goal of capturing "a coherent signal of acidification".

**Measurement uncertainty**: The authors find an offset between in-situ and comparison measurements during the test deployment. The linear regression line in Fig.2 reveals that this offset is pH-dependent and ranges between 0.1 and 0.18 pH units. The reported discrepancy is significantly larger than the Global Ocean Acidification Observing Network (GOA-ON) "weather" and "climate" goals for pH measurements, claiming uncertainties of +/- 0.02 and +/- 0.003, respectively (Newton et al., 2015). It also exceeds the accuracy estimate stated by the authors (+/- 0.003). Any attribution, explanation

or correction of this offset is missing. Central shortcomings in this respect are the lack of any methodological information about the spectrophotometric comparison measurements performed in the laboratory on discrete samples (see specific comments) and the lack of raw data, which prevents an assessment of the source of error. It remains thus unclear, whether the sensor or the laboratory (or both) measurements fail to achieve the required accuracy. As a consequence, any trend estimates derived from comparison of this data set to future observations is at least highly questionable, if not misleading.

**Data processing**: The applied "quality control" procedures appear inappropriate. In a first step, pH date outside a predefined range (7.5-8.25, l.112) are removed. This range is narrower than the stated application range of the method (7-9, l.43). In a second step, pH data that deviate by more than two times the standard deviation from a calculated 12.5-hr running average are removed. Both steps risk to discard environmentally relevant pH variability and would only be justified if a reliable proof is given, that those procedures are appropriate to separate instrumental noise from environmental variability.

**Noisy data**: The authors argue that the higher noise in deployments 1 and 2 is caused by suspended sediments. If this is the case, than the recorded data do not represent environmental pH, are not meaningful for further interpretation, and should thus be removed from the database and manuscript.

**Completeness of data**: The authors mention that the pH data presented here were recorded in the framework of the project A.RIOS which aims at establishing an observation network of ocean acidification. However, the manuscript lacks any information about additional observations available from this network. This seems to conflict with the ESSD guideline which states that "a data set or collection must not be split intentionally".

**In summary, it is suggested to**:

- revise and explain laboratory comparison measurements scrupulously, identify the reason for the observed offset and check whether any additional corrections must be applied (see specific comments)

- include raw data from laboratory and sensor measurements in data set in order to enable a re-processing of the data

- remove deployment 1 and 2 from database

- publish data from deployment 3 and 4 only if offset from comparison measurements can be explained and corrected

- ideally, combine data presented here with additional future pH data and other environmental data gained through observation network, as this would increase the usefulness of the data set

- resubmit manuscript when the combined data set allows trend analysis or assessment of drivers of variability

**3   Specific comments**

**Material and Methods**

l. 41-42: How were instrument specifications (accuracy, drift behavior, precision) determined?

l. 43: How was the application range (7-9) specified?

l. 51: Please specify what you mean by "instrument reasonably simple to operate"

l. 55: What are "standard sampling methods"? Please specify or provide unambiguous reference.

l. 56: Where, when and at which depth was the instrument deployed? What were the environmental conditions during this deployment? All laboratory and field test data from the initial test deployment should be made accessible.

l. 58: Any information about spectrophotometric measurements on discrete samples is missing. This is very critical, because it makes it impossible to attribute the observed offset to the sensor data. The following information must be included:

- What kind of equipment was used?

- Were dye impurities corrected or was a purified dye used? (Douglas and Byrne, 2017; Liu et al., 2011)

- How was the dye pH-perturbation corrected (Carter et al., 2013; Hammer et al., 2014)

- Which dye characterization was used to calculate pH from the absorbance ratio R? This is neither referenced for laboratory nor in-situ measurements (Clayton and Byrne, 1993; Liu et al., 2011; Müller and Rehder, 2018).

- At which temperature were laboratory measurements performed? How was the temperature adjusted to match the in-situ measurements? At which temperature are pH results reported in general?

- Please include raw data (R-value, S and T) for all field and laboratory measurements, in order to enable a re-processing of the observations

l. 61: What exactly do you mean by "Although the relationship was not one of identity, it was consistent"? The slope of the linear regression (0.651, Fig. 2) indicates that the pH offset is not constant, but rather a function of the absolute pH.

l. 61: Why were the authors "encouraged by the results"? The offset is 1-2 orders of magnitude larger than the expected accuracy and has systematic pH-dependence. Please note that neither GOA-ON "weather" nor "climate" criteria are fulfilled.

l. 89: If "the resuspension of sediment" is the cause for the noise observed in deployment 1 and 2, than those data need to be removed.

l. 97: Which temperature record is shown in Fig. 5?

l. 106: What do you mean by "we aligned the temperature, conductivity, and pressure data with the pH time-series"? Does this refer to some kind of interpolation?

l. 109: Did you apply any salinity correction to deployments 1 and 2? If not, please state this explicitly.

l. 111-115: The procedures described here are not quality control measured, but rather an attempt to remove noisy data. It must be clearly argued why data outside the pH-range 7.5 - 8.25 and 2x the SD from the running average are removed.

l. 114: It remains unclear, whether "clean" time series refers to the recorded data or the rolling average. Please specify.

**Results**

l. 120: Linear regression of pH against time does not seem to be a reasonable analysis on those short time scales. Is there any reason to expect a linear change of pH over the course of the deployments? If not, please remove the regression lines.

l. 125: Please present correlation plots of pH with temperature and salinity if you intend to discuss those.

**Figures**

Fig.2: Plot and discuss also the pH offset as a function of pH.

Fig.4: Individual data points can not be identified in the line plot. Please increase x-axis

to full page width and include points. This would allow to identify patterns in the data, such as in Fig. 1 of this review generated online on the PANGEA website.

Fig.5: Remove linear regression.

**Data sets stored at Pangea**

Raw data that would allow to re-process the data (for example if new dye characterizations become available) are missing. Please include those in the data set.

Deployment location appears on Corse on the build in map (presumably due to a wrong sign of longitude values). See Fig. 2 of this review and please correct.

**4   Technical corrections**

Grammar and wording were not reviewed due to severe methodological limitations and concerns of scope, which need to be addressed first. In general the presentation quality is fair, but in some parts poor grammar and wording make it hard to understand the meaning unambiguously.

The mentioned R code written to perform data processing, quality control and visualization is not made accessible.

**5   References**

Carter, B. R., Radich, J. A., Doyle, H. L. and Dickson, A. G.: An automated system for spectrophotometric seawater pH measurements: Automated spectrophotometric pH measurement, Limnol. Oceanogr. Methods, 11(1), 16–27, doi:10.4319/lom.2013.11.16, 2013.

ESSDD
[Figure]

Clayton, T. D. and Byrne, R. H.: Spectrophotometric seawater pH measurements: total hydrogen ion concentration scale calibration of m-cresol purple and at-sea results, Deep Sea Res. Part Oceanogr. Res. Pap., 40(10), 2115–2129, doi:10.1016/0967-0637(93)90048-8, 1993.

Douglas, N. K. and Byrne, R. H.: Achieving accurate spectrophotometric pH measurements using unpurified meta-cresol purple, Mar. Chem., 190, 66–72, doi:10.1016/j.marchem.2017.02.004, 2017.

Hammer, K., Schneider, B., Kuliński, K. and Schulz-Bull, D. E.: Precision and accuracy of spectrophotometric pH measurements at environmental conditions in the Baltic Sea, Estuar. Coast. Shelf Sci., 146, 24–32, doi:10.1016/j.ecss.2014.05.003, 2014.

Liu, X., Patsavas, M. C. and Byrne, R. H.: Purification and Characterization of meta-Cresol Purple for Spectrophotometric Seawater pH Measurements, Environ. Sci. Technol., 45(11), 4862–4868, doi:10.1021/es200665d, 2011.

Müller, J. D. and Rehder, G.: Metrology of pH Measurements in Brackish Waters—-Part 2: Experimental Characterization of Purified meta-Cresol Purple for Spectrophotometric pHT Measurements, Front. Mar. Sci., 5, 177, doi:10.3389/fmars.2018.00177, 2018.

Newton, J., Feely, R., Jewett, E., Williamson, P. and Mathis, J.: Global Ocean Acidification Observing Network: Requirements and Governance Plan, , 61, 2015.
* * *
[Figure]

**Fig. 1.** pH time series visualization from PANGAEA (deployment 1, pH plotted over row number)

[Figure]

**Varela, Ramiro A; Herrera, Juan Luis; González, Jose; Rosón, G; Pérez, Fiz F (2019):**
Deployment 2: pH and temperature measurements in the bottom layer of the Ria de Vigo
(NW Spain). *PANGAEA*, https://doi.org/10.1594/PANGAEA.909930,

*In:* Varela, RA et al. (2019): First automatic pH measurements in the bottom layer of the
Ria de Vigo (NW Spain). *PANGAEA*, https://doi.org/10.1594/PANGAEA.909933

Always quote above citation when using data! You can download the citation in several formats below.

RIS Citation | BibTeX Citation | Copy Citation | Facebook | Twitter | Show Map | Google Earth

**Fig. 2.** Sampling location as shown on PANGAEA

---

## Author Comment (AC1) · 19 Jun 2020

Dear Michele,

Thanks for your review. We agree that a more in-depth analysis of the data variability would significantly improve the paper, an aspect that could be addressed better with more data. That, together with your concerns regarding the noise of the two first deployment (concerns shared by our other reviewer), has led us to delay the resubmission of our manuscript until better quality and longer series are available.

Bests,

Juan

---

## Author Comment (AC2) · 19 Jun 2020

Dear Jens,

Thank you very much for your detailed review of our article. You pointed out several weaknesses of our manuscript, and we agree with many of them. Although we could address many of your concerns in a reviewed version of the paper, we can hardly dispute the points regarding the completeness and noise of our data.

By submitting this paper, we intended to share our experience with this measurement method, and then update the PANGAEA repository later as new data was acquired. Still, it is probably a better idea to wait until that new data is available and then resubmit the manuscript with a combined dataset.

Bests,

Juan